# Controlled Carboxylic Acid-Functionalized Silicon Nitride Surfaces through Supersonic Molecular Beam Deposition

**DOI:** 10.3390/ma16155390

**Published:** 2023-07-31

**Authors:** Marco V. Nardi, Melanie Timpel, Laura Pasquardini, Tullio Toccoli, Marina Scarpa, Roberto Verucchi

**Affiliations:** 1Institute of Materials for Electronics and Magnetism (IMEM-CNR), Trento Unit c/o Fondazione Bruno Kessler, Via alla Cascata 56/C, 38123 Trento, Italy; melanie.timpel@imem.cnr.it (M.T.); tullio.toccoli@imem.cnr.it (T.T.); 2Fondazione Bruno Kessler, Via Sommarive 18, 38123 Trento, Italy; l.pasquardini@gmail.com; 3Dipartimento di Fisica, Nanoscience Laboratory, Via Sommarive, 14, 38123 Trento, Italy; marina.scarpa@unitn.it

**Keywords:** functionalization, photoelectron spectroscopy, silicon nitride, naphthalene

## Abstract

The functionalization of inorganic surfaces by organic functional molecules is a viable and promising method towards the realization of novel classes of biosensing devices. The proper comprehension of the chemical properties of the interface, as well as of the number of active binding sites for bioreceptor molecules are characteristics that will determine the interaction of the sensor with the analyte, and thus its final efficiency. We present a new and reliable surface functionalization route based on supersonic molecular beam deposition (SuMBD) using 2,6-naphthalene dicarboxylic acid as a bi-functional molecular linker on the chemically inert silicon nitride surface to further allow for stable and homogeneous attachment of biomolecules. The kinetically activated binding of the molecular layer to silicon nitride and the growth as a function of deposition time was studied by X-ray photoelectron spectroscopy, and the properties of films with different thicknesses were investigated by optical and vibrational spectroscopies. After subsequent attachment of a biological probe, fluorescence analysis was used to estimate the molecular layer’s surface density. The successful functionalization of silicon nitride surface via SuMBD and the detailed growth and interface analysis paves the way for reliably attaching bioreceptor molecules onto the silicon nitride surface.

## 1. Introduction

Over the past decades, biosensing technologies rapidly emerged as new diagnostic tools [1,2], since biological targets and processes are fundamental in a wide range of applications, such as human healthcare, agri-food, environmental science, and security networks. A biosensor is basically a hybrid device, where a biological element which recognizes the analyte is connected to a physicochemical transducer. Besides silicon-based sensing platforms, silicon nitride has been shown to be a suitable transducer material for optical [3,4,5] and electrochemical [6] biosensing technologies, mainly due to its superior optical properties and chemical inertness, respectively.

In the biomedical field, silicon nitride has been proven to be biocompatible, with a favorable osteogenic promotion ability both in vitro and in vivo and also antibacterial effectiveness [7]. It can be used to build implantable sensor devices [8]. In the biosensor field, a silicon nitride-based transducer is used for the detection of tumor necrosis factor alpha (TNF-α), a potential biomarker detected in both blood and saliva in the acute stage of inflammation [6]. Being a CMOS-compatible material with a lower refractive index with respect to silicon, silicon nitride finds several applications in photonic devices. For instance, an optofluidic device based on a silicon nitride microring resonator has been used to detect the conversion of a substrate by an enzyme in the visible range [9,10]. Further visible light applications of silicon nitride are in optical coherence tomography devices, as multi-spectral light sources to enhance the resolution in microscopy or flow cytometry, and as diagnostic lab-on-a-chip sensors [11].

Several approaches have been applied to induce functional groups on the Si_3_N_4_ surface, such as plasma treatment [12], UV irradiation [13], silanization methods [14,15], and other wet-chemistry approaches [16,17,18]. Monolayer formation by wet chemistry represents the most common and widespread method, mainly due to its reliability. However, these methods can require several chemical treatments to remove organic contaminations and a sequence of chemical reactions to bind the intermediate layer useful for the final immobilization of the bioreceptor (antibodies, aptamers, etc.).

In this work, we demonstrate that using an ultra-high vacuum (UHV) environment coupled with supersonic molecular beam deposition (SuMBD), is a robust, fast, and reliable approach for molecular functionalization of chemically inert silicon nitride surfaces. Inorganic surfaces can be directly sensitized in a controlled ultra-high vacuum (UHV) environment, avoiding surface contamination via chemical treatments (e.g., using organic solvents) which are usually applied during conventional wet-chemical functionalization approaches [19], while the SuMBD method with the possibility to increase the kinetic energy of the impinging molecules promotes their reaction with the surface. This method is based on the free-jet expansion in vacuum of a light carrier gas seeded by organic precursors [20] and gives the possibility to widely control the energetic state of the impinging molecules in terms of kinetic energy, momentum, and internal degree of freedom by optimizing the initial working conditions such as the source nozzle dimensions, temperature, pressure of the reservoir, etc.

Seeding the vapor of a molecule in a lighter carrier gas, e.g., helium (He), makes it possible to control the final state of the seeded molecules [20]. During the supersonic expansion, the internal degrees of freedom of the molecules undergo a strong cooling process while the kinetic energy can be increased, namely, from a few hundreds of meV up to several eV by varying the degree of seeding [20]. By tuning the SuMBD parameters, it is possible to grow ordered organic thin films [21,22] and to control the activation of chemical bonds at intrinsically inert surfaces [23].

In the present study, 2,6-naphthalene dicarboxylic acid (NDCA) was chosen as a bi-functional linker molecule, consisting of a naphthalene core and two opposite carboxylic (-COOH) end groups (see inlet in Figure 1), and exhibiting ideal chemical and thermal stability for UHV deposition. It was found that covalent bonds form between the Si_3_N_4_ surface and one of the NDCA’s carboxylic groups, leading to an interlayer with the other carboxylic group standing upright on the surface, which can be further used for anchoring of biomolecules.

NDCA films with different thicknesses as well as the COOH-functionalized Si_3_N_4_ interface were analyzed via X-ray photoelectron spectroscopy (XPS) to understand the SuMBD growth behavior of NDCA on Si_3_N_4_. Fourier Transform infrared spectroscopy (FTIR) and µ-FTIR imaging allowed to probe the surface density of the functionalized sites available for biomolecule anchoring. Moreover, a standard fluorescence spectroscopic analysis was performed to quantify the surface density of the fluorescent aptamer immobilized on Si_3_N_4_.

## 2. Materials and Methods

### 2.1. Materials

The molecule 2,6-naphthalene dicarboxylic acid (NDCA) used for surface functionalization, a 0.1 M phosphate buffer solution (at pH 7.4), and hydrofluoric acid (HF) for Si_3_N_4_ etching were purchased from Sigma Aldrich (Milan, Italy). The Si_3_N_4_ substrates were fabricated by low-pressure chemical vapour deposition (LPCVD) on Si wafers [24,25] with thicknesses of about 60 nm (areas of 2 cm^2^). A thrombin binding aptamer (TBA) used for fluorescence labelling was purchased from IDT Integrated DNA Technologies (Leuven, Belgium). The HPLC-purified DNA-sequence of the TBA molecule was 5’-NH_2_-(CH_2_)_12_-GGT TGG TGT GGT TGG-(CH_2_)_3_-fluoresceinammine-3’, hereafter called TBA-FAM. The crosslinkers 1-ethyl-3-(-3-dimethylaminopropyl) carbodiimide hydrochloride (EDC) and water-soluble N-hydroxysuccinimide (sulfo-NHS) as well as Triton^TM^ X-100 (TX-100) were purchased from ThermoFisher Scientific Inc, whereas 6-aminofluorescein (AMF) was purchased from Sigma-Aldrich (Milan, Italy).

### 2.2. NDCA Thin Film Grown by SuMBD

Before SuMBD deposition of NDCA, the Si_3_N_4_ surface was etched in HF solution (0.1 M) for 30 min to remove the residual oxynitride layer and to form an NH-terminated Si_3_N_4_ surface. After etching, the sample was rinsed with deionized water and quickly inserted into the UHV chamber to minimize contaminants from air exposure. Details of the experimental SuMBD set-up were previously described in Ref. [20]. Briefly, the SuMBD beam source is equipped with a nozzle exhibiting a diameter of about 130 µm, and a typical working sublimation temperature of about 170 °C was used. In the present study, the SuMBD working conditions were systematically optimized to improve the surface reaction of the NDCA molecule with the Si_3_N_4_ surface. This was made in an apparatus where an in-line time-of flight mass spectrometer (TOF-MS) is coupled with the SuMBD chamber. The in-line configuration allows to extrapolate the kinetic energy of the molecules from their time of arrival in the mass spectra [20]. Figure 1 shows the kinetic energy of the NDCA molecule by varying the degree of seeding of NDCA in He carrier gas (as detected by TOF-MS). A maximum kinetic energy of the He carrier gas of 6 eV was chosen for the growth of the NDCA films on Si_3_N_4_ to improve the possibility to activate surface reactions. NDCA films with thicknesses of about 10 nm (thin films) and hundreds of nm (bulk) were grown by varying the deposition time between 25 min and 13 h 20 min. After surface functionalization, the samples were further sonicated in aqueous phosphate buffer solution for 10 min to remove physisorbed molecules, exposing the molecular layer covalently bonded to the Si_3_N_4_ surface. 

### 2.3. XPS Analysis

X-ray photoelectron spectroscopy (XPS) was used to detect the presence of chemical species evidencing the formation of covalent bonds. XPS was performed with a non-monochromatized Mg Kα source (emission line at 1253.6 eV). The photoelectron signal was detected with a VSW HSA100 hemispherical analyzer (PSP Vacuum Technology Ltd., Cheshire, UK) equipped with a PSP electronic power supply and control. The total energy resolution was about 0.8 eV. The binding energy (BE) scale of the XPS spectra was calibrated using the Au 4f peak at 84.0 eV of an Au reference sample.

### 2.4. UV/Vis Absorption Spectroscopy

The UV/Vis absorption spectra were collected on NDCA films grown on quartz (using the same deposition parameter than for Si_3_N_4_) in a Varian Cary 5000 spectrometer (Agilent Technologies, Santa Clara, CA, USA) in the 500–200 nm range with a scan rate of 150 nm/min and a bandwidth of 2 nm in a double beam configuration.

### 2.5. FTIR Analysis

FTIR analysis and µ-FTIR mapping was performed with a Nicolet iN10 micro-FTIR instrument from Thermo Fischer equipped with a motorized stage. FTIR spectra were collected in the 1000–1750 cm^−1^ range with a resolution of 4 cm^−1^. µ-FTIR mapping was recorded on a sample area of 1.5 mm^2^ at room temperature with a spectral resolution of 16 cm^−1^ and a spatial resolution of <10 μm.

### 2.6. Fluorescence Labelling

Labelling by fluorescence markers carrying an amino group was utilized to evaluate quantitatively the efficiency of the functionalization steps. To this purpose, 6-aminofluorescein was utilized as a low-molecular weight label of the free COOH groups after NDCA functionalization. To evaluate the bioconjugation of the COOH groups to aptamers, a DNA-aptamer sequence carrying an amino chemical group at 5’ end and a fluorescent molecule at 3’ end (TBA-FAM) was used. The aptamer density was measured before and after sonication (for 20 min in water) of the functionalized sample, to consider the possible physical adsorption of DNA sequences on the surface.

To perform the labelling reactions, after extensive washing, the carboxylic groups on NCDA-functionalized Si_3_N_4_ surface were activated using an 8/2 mM EDC/sulpho-NHS molar ratio in MES buffer 50 mM (pH = 5.5) for 3 h in orbital shaking. Then, 1 mM 6-aminofluorscein in phosphate buffer 0.05 M (pH 8) for 2 h or 2.5 µM TBA-FAM in phosphate buffer 0.2 M (pH = 7.5) for 1 h in orbital shaking were added. To unfold the sequence strands of TBA-FAM and get the amino groups available for the immobilization reaction, a thermal treatment at 95 °C for one minute was performed, followed by thermal shock in ice for 10 min. 

The quantification of carboxylic groups was obtained by measuring the fluorescence of the conjugated 6-aminofluorescein after its detachment from the sample surface. To this purpose, the 6-aminofluorescein labelled Si_3_N_4_ sample was incubated in 4% HF containing 1% *v*/*v* of TX-100 for 30 min in an orbital shaker. Then, the solution was brought at pH 9 by NaOH. The fluorescence signal of this solution was measured, and the concentration was calculated with respect to a calibration curve previously acquired under the same experimental conditions.

To evaluate the efficiency of EDC/sulpho-NHS activation on the aptamer binding and rule out the possible contribution of un-specific adsorption, a COOH-functionalized surface was incubated with DNA-aptamer sequence without activation. After incubation, the surfaces were washed in the same buffer and the fluorescence signal was recorded using a fluorescence microscope Leica DMLA (Leica Microsystems, Wetzlar, Germany), equipped with a mercury lamp and fluorescence filter L5 (Leica Microsystems, Wetzlar, Germany). A cooled CCD camera (DFC 420C, Leica Microsystems, Germany) was used to acquire the images, which were analyzed with the software ImageJ [26]. 

## 3. Results

The X-ray photoemission spectroscopy (XPS) chemical analysis of the NDCA-functionalized Si_3_N_4_ surfaces is reported in Figure 2. We found the presence of carbon, oxygen, silicon, and nitrogen-related species, without any fluorine trace. Representative C 1s and O 1s spectra of the NDCA thin film and corresponding sonicated film are shown in Figure 2a,b, respectively, whereas Figure 2c illustrates the (substrate-specific) N 1s core levels of the bare Si_3_N_4_ substrate and the sonicated film. All core level fitting components, their binding energies (BEs), and corresponding assignments are listed in Table 1. The C 1s core level spectrum of the NDCA thin film (lower panel in Figure 2a) is characterized by a main peak (C_A_ component) corresponding to the C bonds of the NCDA’s naphthalene core, and a smaller contribution (C_B_ component) corresponding to the C bonds of the two carboxylic (COOH) end groups [27,28,29]. The C_D_ broad peak is due to a shakeup process of the photoelectrons originated from one (or more) carbon species, typical for π-conjugated molecules [30]. After sonication in phosphate buffer solution, the intensity of the COOH peak is markedly decreased whereas an additional interface component (C_C_ component) appears, related to the NCDA’s COOH group that reacted with Si_3_N_4_.

The O 1s core level spectra shown in Figure 2b are representative for thin and sonicated film, respectively, after subtraction of a small amount of residual oxynitride content after HF treatment of Si_3_N_4_. The thin film (lower panel in Figure 2b) exhibits two components with similar intensity and corresponding to the two O atoms in the carboxylic end groups, i.e., C=O at lower BE (O_A_ component) and C-OH at higher BE (O_B_ component), respectively [27,28]. After sonication, the ratio between the two components markedly changed, i.e., the O_A_ component exhibits a three-times-higher intensity than the O_B_ component. This can be attributed to the fact that the O_A_ component at the interface represents two (unreacted) C=O groups and one C-OH group that has reacted with the Si_3_N_4_ surface, whereas the interfacial O_B_ component corresponds to the NDCA’s unreacted C-OH group.

The reactivity of NDCA molecules on Si_3_N_4_ grown via SuMBD is further evidenced by comparison of the substrate-specific N 1s core level spectra (Figure 2c). The NDCA- functionalized Si_3_N_4_ surface (upper panel in Figure 2c) exhibits a weak additional component (N_B_) with respect to the etched Si_3_N_4_ substrate (upper panel in Figure 2c) [31], corresponding to the surface nitrogen component that reacted with the NDCA molecule.

The growth of NDCA thin films on etched Si_3_N_4_ substrates via SuMBD was further evaluated by the relative attenuation and increase (in %) in the substrate-specific (Si 2p and N 1s) and molecule-specific (C 1s) core levels (Figure 3), respectively, after different deposition times varying from 20 min (T1) to 13 h 20 min (T8). As can be seen in Figure 3, the experimental curves were compared with the calculated attenuation/increase of the core levels for thin films grown via layer-by-layer (or Frank–van der Merwe) mode [32].

As can be seen from Figure 3a,b, the experimental curves markedly deviate from theoretical layer growth. They are clearly constituted by two different growth regimes (as separated by dashed lines in Figure 3). The first regime develops between T1 and T4 and exhibits linear behavior. A clear change in the behavior close to T4 is visible, and a second regime appears, where the signal of substrate and overlayer asymptotically approaches full attenuation and increase, respectively.

Optical absorption (UV/Vis) and FTIR spectroscopy were performed on NDCA-functionalized substrates (quartz and Si_3_N_4_, respectively) to investigate the impact of sonication on the properties of the molecular layer (see Figure 4). The absorption spectra of both thick and sonicated NDCA film (Figure 4a) exhibit the same line shape with pronounced features at 220, 250 and 310 nm, and an onset at 380 nm. The detailed analysis of the optical absorption spectra is beyond the scope of this work.

Figure 4b shows FTIR spectra and corresponding band assignments of thick, thin, and sonicated NDCA film on Si_3_N_4_. Typical vibrational features of the molecule are detected for each sample. The FT-IR spectra of both thick and thin films exhibit multiple bands in the region 1700–1100 cm^−1^. The bands in the region 1600–1550 cm^−1^ agree well with literature values for the ring stretching modes of C-C and C=C bonds in the naphthalene core. There are several bands at lower energy (e.g., 1250–1500 cm^−1^) that are related to C-C and C-H bending in the naphthalene core and carboxylic group. Additional bands are visible at ∼1100–1200 cm^−1^ which contain both C-O and C-OH stretching and CCH deformation character [33]. Overall, the findings shown in Figure 4 suggest that sonication treatment does not affect the properties of the NDCA molecular layer.

We further analyzed the molecular distribution and coverage uniformity by monitoring the COOH peak (i.e., C=O stretching) at ∼1690 cm^−1^ in a region of ~1.5 mm^2^, see µ-FTIR maps in Figure 5. Both bulk and thin NDCA films exhibit island-like structures. After sonication, the homogenous area amounts to 99.2% of the total area, indicating the formation of a uniform and stable COOH-functionalized Si_3_N_4_ surface. It should be noted that a small aggregate from the NDCA thick film is still present. It represents less than 1% of the analyzed surface and is probably due to a very high island not completely removed after sonication.

Figure 6 reports the fluorescence signal of the TBA-FAM aptamer immobilized on the NCDA-functionalized surface.

The overall fluorescence intensity is higher on the activated samples, confirming the availability of the carboxylic groups on the NDCA-functionalized Si_3_N_4_ surface. After sonication, the surface shows very similar homogeneity (Figure 6b), without significantly losing the fluorescence intensity (as evidenced by the measured fluorescence intensity values Figure 6c). The DNA-aptamer is therefore successfully immobilized on the NCDA-functionalized Si_3_N_4_ surface. 

The presence of free carboxylic groups was quantitatively confirmed by labelling these groups with 6-aminofluorescein, according to the protocol reported in the methods section. In particular, the surface density of carboxylic groups available as binding sites for bioreceptors on the Si_3_N_4_ surface was found to be (4.8 ± 1.4) × 10^12^ COOH groups/cm^2^.

## 4. Discussion

In the present study, we optimized the SuMBD growth to kinetically activate NDCA molecules that can be grafted as a COOH-terminated linker to Si_3_N_4_ surfaces (Figure 1). The analyses of films with different thicknesses as well as sonicated NDCA films (Figure 2, Figure 4 and Figure 5) indicate that sonication of the samples in aqueous buffer solution removes the weakly bound molecules, exposing a homogenous NDCA monolayer covalently bound to the Si_3_N_4_ surface while keeping the layer’s molecule structure intact. The monolayer formation most likely occurred through C–N covalent bonds between one carboxylic group of NDCA and the nitrogen atoms of the Si_3_N_4_ surface. The role of other chemical groups on the surface (e.g., -OH) cannot be entirely excluded. However, the chemical bond at the organic/inorganic interface is strong, most likely covalent, as supported by our findings that all weakly bounded species are removed by sonication. This in turn leads to an upward-oriented COOH group available for further attachment of biomolecules. The interlayer formation (SuMBD and sonication in water/aqueous buffer solution) is achievable in about 30 min without the use of organic solvents, which makes the entire process comparably faster and simpler with respect to conventional wet-chemical approaches to functionalize oxide-free silicon-based surfaces.

The growth behavior of NDCA films on Si_3_N_4_ was then systematically investigated by XPS analysis. The evolution of the XPS peak area for the substrate-specific core levels (i.e., Si 2p and N 1s) and the overlayer (i.e., C 1s) as a function of deposition time (see Figure 3) clearly indicate a characteristic Stranski–Krastanov layer-plus-island growth where the nucleation of islands begins after the complete formation and coalescence of the first two–three layers (i.e., at point T4 in Figure 3). The development of the NDCA film morphology with high-aspect-ratio islands and progressive increase in the film thickness is also confirmed by µ-FTIR mapping. The overall findings are schematically illustrated in Figure 5 (upper panel). 

The absorption spectra in Figure 4 verify the suitability of NDCA as functionalizing molecule for Si_3_N_4_-based optical biosensors. The low absorption of the molecular layers (i.e., thick and sonicated films) in the UV-Vis range guarantees a negligible absorption of the light during excitation or fluorescence for aptamers/protein detection in optical biosensors. Furthermore, consistently with the very low absorption of NDCA, a negligible auto-fluorescence is expected. 

To demonstrate proof-of-concept for biosensing, we attached different biomolecules (TBA-FAM and AMF) to the carboxylic binding sites at the Si_3_N_4_ surface by coupling the carboxylic acid to the amine group of the biomolecule via carbodiimide chemistry. This kind of chemistry is widely used as a conjugation strategy to immobilize biomolecules on carboxylic activated surfaces [19,34]. The carboxylic binding sites accessible to bioreceptor molecules depend on the deposition method and on the type of substrate. Liu and co-workers, for instance, compared the surface densities of COOH groups on gold substrates using molecules that have different spatial conformations [35]. They showed that the steric hindrance of the molecule carrying the carboxylic moiety changes the final surface density of the COOH groups by four times. On a silicon nitride substrate, a density in the order of 10^13^ COOH groups/cm^2^ has been reported using wet-chemical functionalization methods [36], which is in good agreement with the density of carboxylic binding sites measured in the present work. Other approaches have been used to improve the density of carboxylic groups on silicon and silicon nitride surfaces as reported by Cattaruzza et al. [16], obtaining higher surface densities. It should be noted that in the biosensor field, high surface densities are not always desirable because steric hindrance can occur in such densely packed interlayers, so that bioreceptors such as aptamers or antibodies might not have enough space to reach the proper three-dimensional conformation.

## 5. Conclusions

In this work, we demonstrated the possibility of functionalizing the Si_3_N_4_ technological surface with 2,6-naphthalene dicarboxylic acid (NDCA) through the supersonic molecular beam deposition (SuMBD), a technique working in vacuum with a high level of purity. We covalently bonded the organic molecule to the nitride surface by means of a COOH–NH interaction, leaving the second carboxylic group available for further linking. We successfully anchored two biomolecules (TBA-FAM and AMF) via carbodiimide chemistry, paving the way for the realization of optofluidic devices based on silicon nitride for the sensing of biomolecules.

The direct comparison of wet-chemistry and SuMBD experiments was beyond the scope of this research. However, we demonstrated that the advantageous properties offered by the SuMBD approach for Si_3_N_4_ functionalization outweigh the well-known and conventional additional processes of wet-chemical functionalization to form C–N bonded interlayers. It is predicted that SuMBD will represent an alternative route for surface functionalization and contribute to the development of future silicon nitride-based biosensors.

## Figures and Tables

**Figure 1 materials-16-05390-f001:**
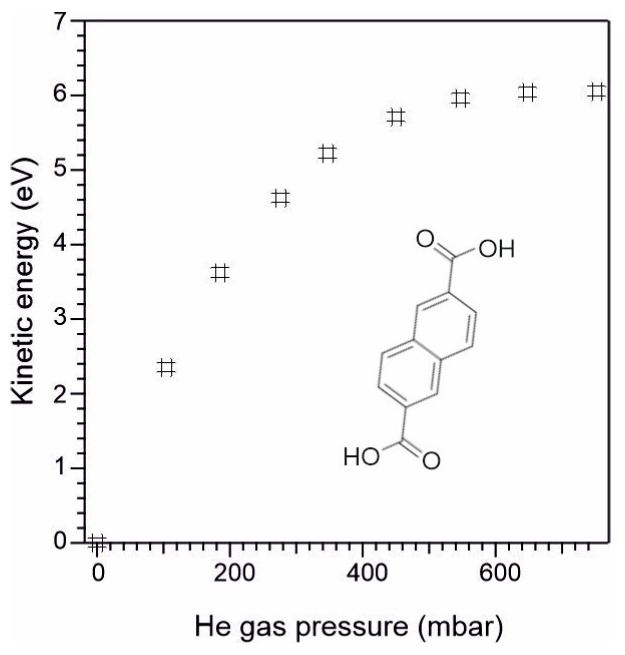
Kinetic energy (measured via TOF-MS) of 2,6-naphthalene dicarboxylic acid (NDCA) molecules as a function of He carrier gas pressure during the SuMBD process. The inset shows the structure of the NDCA molecule used to functionalize the silicon nitride surface.

**Figure 2 materials-16-05390-f002:**
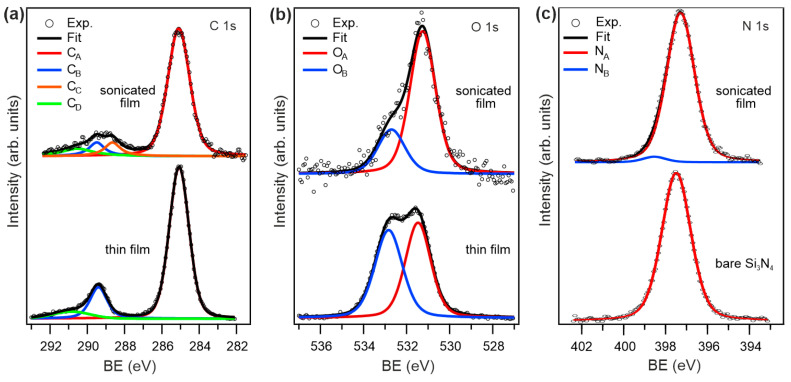
(**a**) C 1s and (**b**) O 1s core-level spectra of NDCA thin film (**lower panel**) grown via SuMBD on Si_3_N_4_ and corresponding sonicated film (**upper panel**); (**c**) N 1s core-level spectra of bare Si_3_N_4_ substrate (**lower panel**) and sonicated NDCA film (**upper panel**).

**Figure 3 materials-16-05390-f003:**
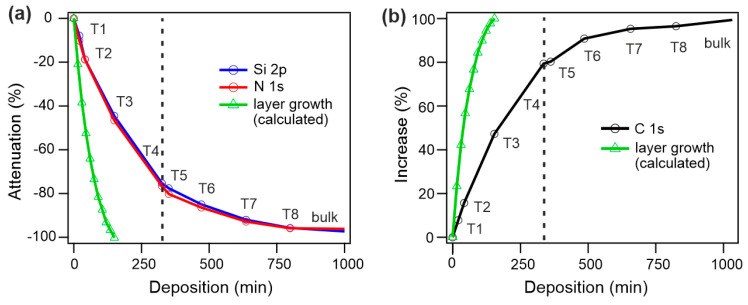
Growth of NDCA on Si_3_N_4_ as a function of deposition time T1–T8: (**a**) attenuation of S 2p and N 1s core levels and (**b**) increase in C 1s core level. The green curves show the theoretical attenuation/increase as expected for thin films grown by Frank–van der Merwe (layer–by–layer) mode.

**Figure 4 materials-16-05390-f004:**
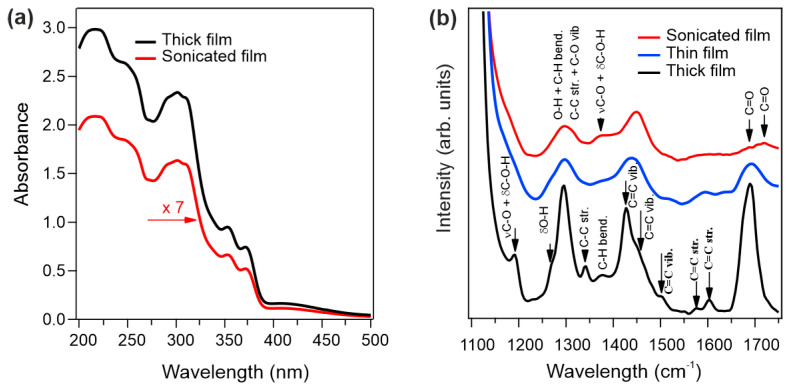
(**a**) UV/Vis absorption spectra of thick and sonicated NDCA film grown on quartz; (**b**) FTIR spectra of thick, thin, and sonicated NDCA film on Si_3_N_4_.

**Figure 5 materials-16-05390-f005:**
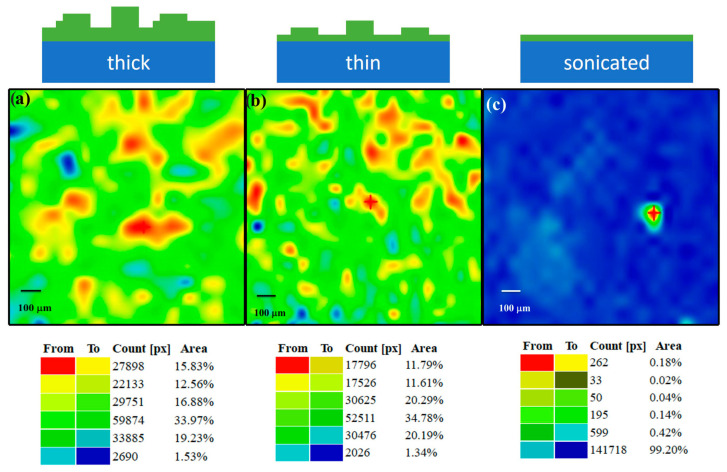
µ-FTIR distribution maps showing the COOH peak intensity of (**a**) thick, (**b**) thin, and (**c**) sonicated NDCA film on Si_3_N_4_. The upper panel schematically illustrates the corresponding film morphology, whereas the lower panel reports the intensity distribution (area in %) of the µ-FTIR maps.

**Figure 6 materials-16-05390-f006:**
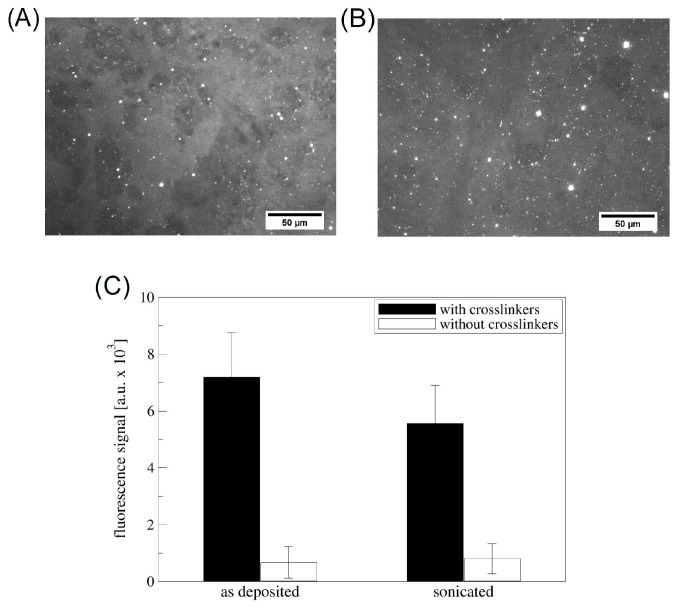
(**A**,**B**) Fluorescence microscopy images of the TBA-FAM distribution on the (**A**) thin and (**B**) sonicated NDCA film; both surfaces have been activated with EDC/sNHS chemistry (with crosslinkers); (**C**) fluorescence intensity values (in greyscale) corresponding to the mean values of six areas on two different NDCA-functionalized Si_3_N_4_ surfaces activated with EDC/sNHS chemistry (with crosslinkers) and not activated (without crosslinkers); the error bars represent the standard deviation.

**Table 1 materials-16-05390-t001:** XPS core level fitting components, their corresponding Bes, and assignments for the thin film and sonicated film as indicated in Figure 2.

**Core Level**	**Label**	**BE (eV)** **Thin Film/Sonicated**	**Assignment**
C 1s	C_A_	285.1/285.1	naphthalene ring
	C_B_	289.4/289.6	COOH
	C_C_	-/288.6	COOH reacted with Si_3_N_4_
	C_D_	290.6/290.8	satellite
O 1s	O_A_	531.5/531.2	C=O + C-OH reacted with Si_3_N_4_
	O_B_	532.8/532.7	C-OH
N 1s	N_A_	397.5/397.4	N in Si_3_N_4_
	N_B_	-/388.6	N reacted with NDCA

## Data Availability

The data presented in this study are available on request from the corresponding author.

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
