# Peer review of "Controlled Carboxylic Acid-Functionalized Silicon Nitride Surfaces through Supersonic Molecular Beam Deposition"

_materials, 2023, doi:10.3390/ma16155390_

Round 1
Reviewer 1 Report
This paper reports a surface functionalization of silicon nitride with supersonic molecular beam deposition. The results show successful deposition of NCDA on the silicon nitride surface and further decoration with biomolecules. It proposes an interesting strategy for immobilizing biomolecules on solid surfaces. However, several points should be discussed before publication.
1. A more detailed discussion on the surface chemistry between silicon nitride and NCDA is required. The authors claimed NH-terminated silicon nitride, however, treatment of silicon nitride with HF can offer other various surfaces such as -OH, -NH2, -SiF, -SiF2, etc.. Furthermore, carboxylic acid or carboxylate can anchor with those units in various ways: H-bonding, electrostatic binding, monodentate, bidentate chelating, bidentate bridging, amide formation, etc..
2. Line 264: It is difficult to find the difference between Figure 6a and 6b. Can the authors quantify the homogeneity?
3. The TBA-FAM treated samples without NCDA show quite an amount of fluorescence (Figure 6c). What would be the reason?
4. The authors emphasize that SuMBD has advantages over wet deposition methods. Important control experiments comparing SuMBD and wet deposition are missing in this research.
Author Response
We thank the reviewer for their valuable comments and suggestions, and for appreciating our work. We addressed each single point as follows, modifying the paper text according to the line number in the revised version:
Reviewer 1
This paper reports a surface functionalization of silicon nitride with supersonic molecular beam deposition. The results show successful deposition of NCDA on the silicon nitride surface and further decoration with biomolecules. It proposes an interesting strategy for immobilizing biomolecules on solid surfaces. However, several points should be discussed before publication.
- A more detailed discussion on the surface chemistry between silicon nitride and NCDA is required. The authors claimed NH-terminated silicon nitride, however, treatment of silicon nitride with HF can offer other various surfaces such as -OH, -NH2, -SiF, -SiF2, etc.. Furthermore, carboxylic acid or carboxylate can anchor with those units in various ways: H-bonding, electrostatic binding, monodentate, bidentate chelating, bidentate bridging, amide formation, etc..
Authors’ response: The HF treated silicon nitride can indeed show different types of groups, as reviewer 1 mentioned. The most widely and commonly accepted one is the -NH termination, but the absence of other groups cannot be ruled out. However, from XPS analysis, we did not find the presence of fluorine. Thus, fluorine-related species are not present on our treated surface.
To make our findings clearer, we added the following sentence (line 189):
“We found the presence of carbon, oxygen, silicon, and nitrogen species without any fluorine trace.”
The presence of an additional component in the N1s core level (blue component in Figure 2c) suggests the involvement of nitrogen surface atoms at the NDCA/Si3N4 interface, but the role of other -OH groups cannot be excluded.
Concerning the type of chemical interaction, we assume the formation of strong covalent bonds at the organic/inorganic interface, where one of the molecule’s ‑COH group and an -NH group at the surface is involved. The strong interaction is proven by the fact that the film remains after the sonication treatment. The successful anchoring of different biomolecules (TBA-FAM and AMF) also suggests that only one carboxylic group is involved in the interface bonding, leaving the other one free for other functionalization processes. Nevertheless, even if we rule out several hypotheses, we agree with the reviewer about the possible presence of other bonded species. We modified our statement (lines 298-303):
“The monolayer formation most likely occurred through C–N covalent bonds between one carboxylic group of NDCA and the nitrogen atoms of the Si3N4 surface. The role of other chemical groups on the surface (e.g., -OH) cannot be entirely excluded. However, the chemical bond at the organic/inorganic interface is strong, most likely covalent, as supported by our findings that all weakly bounded species are removed by sonication.”
- Line 264: It is difficult to find the difference between Figure 6a and 6b. Can the authors quantify the homogeneity?
Authors’ response: The differences between thin (Figure 6a) and (Figure 6b) sonicated NDCA films are shown in Figure 6c, where the mean values of fluorescence intensity are reported (acquired from 6 different areas on two samples) with respective standard deviations. The fluorescence intensity is about 20% less on the sonicated sample, but the homogeneity is very similar, as seen in Figure 6 a, and b. According to the reviewer’s comment, we modified the sentence (line 283-284):
“After sonication, the surface shows very similar homogeneity (Fig. 6b)”.
- The TBA-FAM treated samples without NCDA show quite an amount of fluorescence (Figure 6c). What would be the reason?
Authors’ response: We thank the reviewer for highlighting an unclear figure caption. Figure 6c reports the TBA-FAM distribution on thin and sonicated films of NDCA-functionalized samples with or without the crosslinkers (EDC/sNHS). In both cases, the NDCA film is present on the silicon nitride surface and promotes weak aspecific adsorption when no crosslinkers are used (without). To make it clearer, we modified the Figure 6 caption (lines 276-281):
Figure 6. (a,b) Fluorescence microscopy images of the TBA-FAM distribution on the (a) thin and (b) sonicated NDCA film; both surfaces have been activated with EDC/sNHS chemistry (with crosslinkers); (c) fluorescence intensity values (in greyscale) corresponding to the mean values of six areas on two different NDCA-functionalized Si3N4 surfaces activated with EDC/sNHS chemistry (with crosslinkers) and not activated (without crosslinkers); the error bars represent the standard deviation.
- The authors emphasize that SuMBD has advantages over wet deposition methods. Important control experiments comparing SuMBD and wet deposition are missing in this research.
Authors’ response: SuMBD leads to the use of high-purity species (i.e., after sublimation in vacuum) in a very clean environment (i.e., in ultra-high vacuum), with an overall preparation time that is lower than in typical wet chemical approaches and without the use of organic solvents. We compared our results with already published papers because the direct comparison with specific experiments was out of the scope of our research. We agree on the importance of the reviewer’s comment and added the following sentence in the new chapter “Conclusions” (line 356):
“The direct comparison of wet-chemistry and SuMBD experiments was beyond the scope of this research.”
Reviewer 2 Report
The article presented by the authors is very well discussed, and shows important results for the area of knowledge. However, the manuscript presents few points that should be clarified and explained to final decision.
1) XPS results are well discussed, but only one reference is cited (ref. 22) to support the discussions. If possible, cite new references.
2) Page 6, line 225: UV-vis results are not discussed. Discuss the bands in Figure 4.a and provide references.
3) The text "It can be concluded that the advantageous properties offered by the SuMBD approach for Si3N4 functionalization outweigh the additional processes of wet-chemical functionalization to form C–N bonded interlayers. It is predicted that SuMBD will represent an alternative route for surface functionalization and contribute to the development of future silicon nitride-based biosensors." must be included in a separate item: conclusions. And if possible, improve the conclusions of the work.
4) If possible, insert more references throughout the text, to better support the results obtained.
Author Response
We thank the reviewer for their valuable comments and suggestions, and for appreciating our work. We addressed each single point as follows, modifying the paper text according to the line number in the revised version:
Reviewer 2
The article presented by the authors is very well discussed, and shows important results for the area of knowledge. However, the manuscript presents few points that should be clarified and explained to final decision.
1) XPS results are well discussed, but only one reference is cited (ref. 22) to support the discussions. If possible, cite new references.
Authors’ response: We inserted several references to strengthen the discussion, see new references 27-31 in the revised manuscript. Moreover, we discussed the origin of the satellite in C1s core level, adding the following sentence (lines 198-199):
“The CD broad peak is due to a shakeup process of the photoelectrons originated from one (or more) carbon species, typical for π-conjugated molecules [30].”
2) Page 6, line 225: UV-vis results are not discussed. Discuss the bands in Figure 4.a and provide references.
In the present manuscript, the UV-Vis data serves to compare the thick and sonicated films to understand if the treatment with aqueous phosphate buffer leads to changes in the NDCA films. The detailed discussion of the material’s absorbance spectra is beyond the scope of this work. To clarify our aims, we added the following sentence (line 244):
“The detailed analysis of the optical absorption spectra is beyond the scope of this work.”
3) The text "It can be concluded that the advantageous properties offered by the SuMBD approach for Si3N4 functionalization outweigh the additional processes of wet-chemical functionalization to form C–N bonded interlayers. It is predicted that SuMBD will represent an alternative route for surface functionalization and contribute to the development of future silicon nitride-based biosensors." must be included in a separate item: conclusions. And if possible, improve the conclusions of the work.
Authors’ response: Even though a “Conclusions” section is not explicitly required by the journal format, we agree with the reviewer that a dedicated section could better summarize the results we achieved, emphasizing their importance. We removed the final part of the Discussion section and added a paragraph to improve the conclusions (lines 347-361):
“5. Conclusions
In this work, we demonstrated the possibility of functionalizing the Si3N4 technological surface with 2,6-naphthalene dicarboxylic acid (NDCA) through the supersonic molecular beam deposition (SuMBD), a technique working in vacuum with a high level of purity. We covalently bonded the organic molecule to the nitride surface by means of a COOH-NH interaction, leaving the second carboxylic group available for further linking. We successfully anchored two biomolecules (TBA-FAM and AMF) via carbodiimide chemistry, paving the way for the realization of optofluidic devices based on silicon nitride for the sensing of biomolecules.
The direct comparison of wet chemistry and SuMBD experiments was beyond the scope of this research. However, we demonstrated that the advantageous properties offered by the SuMBD approach for Si3N4 functionalization outweigh the well-known and conventional additional processes of wet-chemical functionalization to form C–N bonded interlayers. It is predicted that SuMBD will represent an alternative route for surface functionalization and contribute to the development of future silicon nitride-based biosensors.”
4) If possible, insert more references throughout the text, to better support the results obtained.
Authors’ response: We added several references in the introduction (Refs. 7-11) and for the discussion of XPS results (Refs. 27-31).
Reviewer 3 Report
In this article, the authors reported a surface functionalization route based on an ultra-high vacuum (UHV) environment coupled with supersonic molecular beam deposition. They optimized the SuMBD growth to activate NDCA molecules grafting as COOH-terminated linker to Si3N4 surfaces. The facile functionalization of the silicon nitride surface via SuMBD, and the detailed growth and interface analysis pave the way for reliably attaching bioreceptor molecules onto the silicon nitride surface.
This article is well-written and offers a versatile method for modifying silicon nitride surfaces, which has great potential for application in biosensors, optical devices, and electrochemistry.
And before publication, some issues need to be addressed.
1. In this article, the author introduced the method for making the silicon nitride surface functional, and how could the functional surface be well used in detail, such as used in biomedical fields, the application should be discussed detailed.
2. Figure 6 shows the fluorescent microscope images of TBA-FAM distribution. But there are only bright field images, the fluorescent images are needed too to illustrate the fluorescent to make Figure 6 (c) reasonable.
3. In Figure 5, after sonication, there is still an ununiform region in the μ -FTIR distribution maps, the reason needs to be discussed and explained.
The English is good.
Author Response
We thank the reviewer for their valuable comments and suggestions, and for appreciating our work. We addressed each single point as follows, modifying the paper text according to the line number in the revised version:
Reviewer 3
In this article, the authors reported a surface functionalization route based on an ultra-high vacuum (UHV) environment coupled with supersonic molecular beam deposition. They optimized the SuMBD growth to activate NDCA molecules grafting as COOH-terminated linker to Si3N4 surfaces. The facile functionalization of the silicon nitride surface via SuMBD, and the detailed growth and interface analysis pave the way for reliably attaching bioreceptor molecules onto the silicon nitride surface.
This article is well-written and offers a versatile method for modifying silicon nitride surfaces, which has great potential for application in biosensors, optical devices, and electrochemistry.
And before publication, some issues need to be addressed.
- In this article, the author introduced the method for making the silicon nitride surface functional, and how could the functional surface be well used in detail, such as used in biomedical fields, the application should be discussed detailed.
Authors’ response: We improved the introduction section by mentioning several examples and citing references (Refs. 7-11) to show the possible applications connected with our approach to silicon nitride. We added the following text (lines 37-48):
“In the biomedical field, silicon nitride has been proven to be biocompatible, with a favorable osteogenic promotion ability both in vitro and in vivo and also antibacterial effectiveness [7]. It can be used to build implantable sensor devices [8]. In the biosensor field, a silicon nitride-based transducer is used for the detection of tumor necrosis factor alpha (TNF-α), a potential biomarker detected in both blood and saliva in the acute stage of inflammation [6]. Being a CMOS-compatible material with a lower refractive index with respect to silicon, silicon nitride finds several applications in photonic devices. For instance, an optofluidic device based on a silicon nitride microring resonator has been used to detect the conversion of a substrate by an enzyme in the visible range [9,10]. Further visible light applications of silicon nitride are in optical coherence tomography devices, as multi-spectral light sources to enhance the resolution in microscopy or flow cytometry, and as diagnostic lab-on-a-chip sensors [11].
- Figure 6 shows the fluorescent microscope images of TBA-FAM distribution. But there are only bright field images, the fluorescent images are needed too to illustrate the fluorescent to make Figure 6 (c) reasonable.
Authors’ response: The images in Figure 6a and 6b do not show bright field images, but the fluorescence intensity in grayscale. To clarify this, we modified the Figure 6 caption (lines 276-281):
Figure 6. (a,b) Fluorescence microscopy images of the TBA-FAM distribution on the (a) thin and (b) sonicated NDCA film; both surfaces have been activated with EDC/sNHS chemistry (with crosslinkers); (c) fluorescence intensity values (in greyscale) corresponding to the mean values of six areas on two different NDCA-functionalized Si3N4 surfaces activated with EDC/sNHS chemistry (with crosslinkers) and not activated (without crosslinkers); the error bars represent the standard deviation.
- In Figure 5, after sonication, there is still an ununiform region in the μ -FTIR distribution maps, the reason needs to be discussed and explained.
Authors’ response: The uniformity on the sample has a percentage of more than 99%, as reported also by the counts below the map. The small aggregate observed in Figure 5c contributes to less than 1% of the figure, meaning that it has no significant influence on the sample homogeneity. We added the following paragraph to improve clarity (lines 263-266):
“It should be noted that a small aggregate from the NDCA thick film is still present. It represents less than 1% of the analyzed surface and is probably due to a very high island not completely removed after sonication.”
Round 2
Reviewer 1 Report
The manuscript is well enhanced, corresponding to the reviewer's comments. I recommend the paper be published in Materials.